# A Fiber Ring Laser Sensor with a Side Polished Evanescent Enhanced Fiber for Highly Sensitive Temperature Measurement

**DOI:** 10.3390/mi12050586

**Published:** 2021-05-20

**Authors:** Weihao Lin, Yibin Liu, Liyang Shao, Mang I. Vai

**Affiliations:** 1Department of Electrical and Electronic Engineering, Southern University of Science and Technology, Shenzhen 518055, China; 11510630@mail.sustech.edu.cn (W.L.); 11853004@mail.sustech.edu.cn (Y.L.); 2Department of Electrical and Computer Engineering, Faculty of Science and Technology, University of Macau, Macau 999078, China

**Keywords:** fiber ring laser, temperature sensor, side-polished fiber, isopropanol

## Abstract

We demonstrate a highly sensitive and practical fiber-based temperature sensor system. The sensor is constructed based on a fiber ring laser (FRL) as well as a side-polished fiber filled with isopropanol. The laser cavity of the sensing part fiber is polished by the wheel fiber polishing system with residual thickness (RT) is selected to detect the temperature in the FRL. Thanks to the high thermo-optic coefficient of isopropanol, the sensitivity of the proposed temperature sensor could be effectively improved by filling isopropanol in the cost-less side polished single mode fiber. Refractive index (RI) of isopropanol changes with the surrounding temperature variation allowing high-sensitivity temperature sensing. Experimental results demonstrate that the side polished fiber can efficiently excite high-order cladding modes which enhance the modular interference increase the interaction between the evanescent wave and the isopropanol. Besides, the results show that the sensitivity can be as high as 2 nm/°C in the temperature range of 25–35 °C.

## 1. Introduction

In recent years, optical fiber sensors have attracted increased interest in aerospace [1], industrial manufacturing [2], medical [3] and other fields, since they offer the advantages of small volume, simple structure, high sensitivity, electromagnetic interference resistance and anti-corrosion properties [4]. In addition, dual-parameter sensors or multi-parameter sensors have been developed and many studies have focused on solving the cross-sensitivity issue [5]. Among them, fiber temperature sensors have been widely explored due to the acceleration of industrial modernization and the rapid development of the information electronics industry. Different kinds of optic fiber temperature sensors have been proposed, i.e., fiber Bragg gratings [6], Mach–Zehnder interferometers (MZIs) [7] and Fabry–Perot interferometers (FPIs) [8]. However, the fabrication and application of these sensing structure enhance the deterioration, resulting in low accuracy of the peak wavelength determination. Wang proposed a response time for an optical microfiber temperature sensor based on the microfiber modal interferometer with sensitivity of −406 pm/°C [9]. Lei et al., demonstrated an optical fiber temperature sensor based on a dual-loop optoelectronic oscillator (OEO) with the Vernier effect by using slightly different lengths of single mode fiber [10]. Besides, there are various optical fiber structures designed for temperature measurement [11,12,13,14,15].

Recently, side polished types of fiber sensors have attracted much attention [16,17,18]. These free devices from the need for additional fiber filters. Pei et al. used a full vector finite element method to design a D-shaped double clad fiber temperature sensor based on surface plasmon resonance (SPR) with sensitivity of −3.635 nm/°C [19]. However, a gold film is needed to coat the fiber, which greatly increases the cost. Jian et al. proposed a novel fiber optic magnetic field sensor based on D-shaped fiber optic modal interferometer filled with a magnetic fluid [20]. The magnetic field sensitivity is 99.68 pm/Oe and the temperature sensitivity is −77.49 pm/°C. The performance of these sensors in terms of temperature is typically limited, the further improvement is desired. In order to meet the requirements of high signal-to-noise ratio and narrow 3-dB bandwidth, optical fiber ring laser sensor systems [21,22,23] have been widely used in recent years. Benefitting from a side-polished optical fiber structure, in 2010 a temperature sensor based on a side-polished fiber (SPF) coupled to a tapered multimode overlay waveguide (MMOW) was proposed by Prerana [24]. Chen et al. designed a novel fiber structure in 2018, in which a coreless side-polished fiber that is wrapped in polydimethylsiloxane [25], whereby a high temperature sensitivity of −0.4409 nm/°C is achieved. Wang el al, demonstrated a surface plasmon resonance (SPR) of a side-polished single mode fiber for temperature sensing thanks to the high thermo-optic and thermal expansion coefficient of the adhesive [26]. In addition, many works describing optical fiber temperature sensors have been reported [19,27,28,29,30].

To address the aforementioned challenges and difficulties, we have designed and fabricated a new sensing system to measure the temperature of the ambient environment by using a side polished evanescent enhanced fiber structure filled with isopropanol in a fiber ring laser system. When the grinding depth is 70.9 μm and 75.5 μm, the temperature sensitivity is measured to be −1.601 nm/°C and −1.921 nm/°C, respectively. The theoretical basis and experimental results are discussed in detail. The OSNR of 50 dB and narrow bandwidth of 0.15 nm have been achieved. Furthermore, the stability of power fluctuation and wavelength shift are also tested and analyzed.

## 2. Working Principle

The experimental setup of the proposed temperature sensor is illustrated in Figure 1. A wavelength division multiplexer (WDM) is used to interconnect the 976 nm pump laser source into a 1.6 m long erbium-doped fiber (EDF) as the gain medium. Afterwards it is connected to an isolator (ISO) to prevent backscattering light. A polarization controller (PC) is inserted in the temperature sensor to regulate the polarization state, the 1% port of the output coupler is used to export light from the FRL cavity for analyzing the temperature change.

Figure 2 illustrates the schematic diagram of the proposed side polished fiber filled with isopropanol configuration, and the inset shows the cross-sectional view of the side polished fiber employed in the experiment. When the incident light transmitting through the single mode fiber with a fundamental mode reach the side-polished region, leading mismatch between core diameter. Hence, the incident light from the incident fiber is split into two parts by the first polishing point, denoted by *I*_core_ and *I*_cavity_. The interference happens when the two beams recombine at the core of the fiber, and the resultant interference light intensity can be mathematically described by the following equation:(1)$$I_{OUT}\left(\lambda \right)=I_{cl}+I_{co}+2\sqrt{I_{cl}I_{co}}cos\phi$$
where $I_{co}$ and $I_{cl}$ delegate the light intensities of beams transmitting through the resid ual core and the side-opened fiber, respectively.

The evanescent wave in cladding mode can be enhanced effectively by polishing the fiber side. In light of the optical transmission process, the refractive index of the cladding modes and core mode rely on the temperature (T) for the thermal-optical coefficient of fiber. The equation could be further simplified as [31]:(2)$$\Phi \;=\frac{2\pi }{\lambda }\left[n_{eff}^{co}\left(\lambda ,T\right)-n_{eff}^{cl,m}\left(\lambda ,T,RI_{ext}\right)\right]L=\frac{2\pi \Delta n_{eff}^{m}\left(\lambda ,T,RI_{ext}\right)L}{\lambda }$$
where $n_{eff}^{co}\left(\lambda ,T\right)$ is the refractive index of the core mode with respect to the temperature change and $n_{eff}^{cl,m}\left(\lambda ,T,RI_{ext}\right)$ (*m* = 1,2,3…) is the refractive index of the *m*-th-cladding modes with respect to the temperature change, $\Delta n_{eff}^{m}\left(\lambda ,T,RI_{ext}\right)L$ is the dissimilarity between the cladding modes and core mode, *L* represents the length of side-polished fiber and *λ* is the center wavelength of input pump source which is 976 nm.

The fringe visibility of the interferogram is calculated as follows:(3)$$V=\frac{2\sqrt{I_{core}I_{cavity}}}{I_{core}+I_{cavity}}$$

Equation (3) shows that the visibility of fringes of interference pattern is mainly determined by the intensity of the core mode and the cavity mode, that is, the smaller the intensity difference between the core mode and the air mode in the grinding and polishing area, the higher the visibility of the fringes of the interference pattern. In the structure of optical fiber based on measurement and polishing, the output region is used as the input coupler of mode excitation, which means that the lateral offset plays an important role in the performance of MZI. In addition, the different grinding depth loss and transmission loss of the core mode and cavity mode also affect the visibility of the fringes. Therefore, we use the COMSOL method to study the waveguide electric field transmission characteristics at 1550 nm wavelength, and the results are shown in Figure 2. It can be seen that some energy transmitted outside the fiber core.

At the end of the introduced fiber, most of the light intensity is concentrated in the SMF core and a small amount of light intensity is distributed in the SMF cladding. If the transverse offset is introduced at the first splice point, the core mode and cavity mode will be excited simultaneously when the incident light is coupled to the polished fiber. The larger the lateral offset is, the more energy is coupled into the cavity mode. This makes it possible to obtain MZI with excellent performance according to Equation (3), which represents a very clear interference spectrum with high signal-to-noise ratio and fringe visibility.

Equation (2) shows that the phase difference between the core mode and the cavity mode strongly depends on their effective RI difference and the length of the interference region. If l is a constant, the temperature sensitivity of MZI can be expressed by Equation (4) [18,32]:(4)$$\frac{d\lambda }{dT}=\frac{\lambda }{n_{eff}^{core}\left(T\right)-n_{eff}^{cavity}\left(T\right)}\left(\frac{dn_{eff}^{core}\left(T\right)}{dT}-\frac{dn_{eff}^{cavity}\left(T\right)}{dT}\right)$$
where $\frac{dn_{eff}^{core}\left(T\right)}{dT}$ and $\frac{dn_{eff}^{core}\left(T\right)}{dT}$ are the thermo-optic coefficients of the RI of fiber core and isopropanol, respectively. The core of the single mode fiber’s thermo-optic coefficient d ~ 8.6 × 10^−6^/°C. While isopropanol with high thermo-optic coefficient of ~−4.5 × 10^−4^/°C is infiltrated into the side polished fiber, the core mode and the isopropanol mode vary quickly and differently when the temperature changes. Besides, since the refractive index value of isopropanol changes with the temperature [21], the effective RI difference between the core mode and the cavity mode will increase with the increment of temperature.

Two different depth of the side-polished fiber were fabricated. The single mode fiber was fabricated by a wheel-based side polishing technique, as shown in Figure 3. where a supercontinuum light source (ASE-C-N, Hoyatek, Shenzhen, China) and an optical spectrum analyzer (OSA, AQ6370D, Yokogawa, Japan) are included. The two ends of the fiber were fixed in the pulley and a 2 g weight is used to straighten the fiber and reduce any vibrations. As shown in Figure 4, the side polished area length is 8 mm and the remnants fiber thickness of the fiber section is about half of the total fiber size, The variations of residual thickness are measured by the microscopy, since the air-cavity is closer to the core, the interaction with core mode and surrounding RI is more sensitivity to temperature, besides, to protect the fiber from being destroyed waists of 75.5 µm and 70.9 µm are selected for temperature sensing as shown in Figure 5.

## 3. Experimental Results and Discussion

Figure 6a shows the output spectrum of the FRL sensing system, whose emission wavelength is ~1550 nm with a threshold pump power of about 400 mW. The emission wavelength of FRL depends on minimum spectral transmission loss, which is mainly determined by the filtering characteristics of the side-polished fiber. Figure 6a shows the output spectrum on the core-off set structure. The SNR of the structure is more than 60 dB, and the 3 dB bandwidth is less than 0.06 nm.

In order to verify the sensing performance of the side-polished fiber optic sensor based on FRL system. The initial response to the surrounding RI was tested, as shown in Figure 6a,b. The wavelength shifts to a short wavelength with refractive index changed from 1.3339 to 1.3865. As shown in Figure 6b, the $R^{2}$ value is 0.978 which indicates that the system has a good linearity to the surrounding environment change and the sensitivity is 46.903 nm/RIU.

With the increase of temperature *t* the transmission spectrum of the side polished fiber in the FRL system shifts accordingly. When the temperature is changed from 25 °C to 31 °C, the spectral response to the different temperatures is shown in Figure 7a,b. Note that the transmission spectrum undergoes a blueshift with the increase of temperature. As shown in Figure 7b, the temperature sensitivity is −1.60 nm/°C. When the diameter of the fiber is 75.5 µm, the relationship between the emission wavelength and the temperature can be fitted linearly (*R*^2^ = 0.998).

The temperature change of the FRL system with polished fiber diameter of 70.9 μm and the spectral change over the 25–35 °C range are shown in Figure 8a,b. As shown in Figure 8b the emission wavelength shifts from 1565.21 nm to 1545.62 nm, corresponding to a temperature sensitivity of −1.96 nm/°C $(R^{2}$ = 0.999).

For the FRL-based sensing system, the stability of the output spectrum is very important. In order to measure the stability of the sensor system, the output power was recorded every 10 min at room temperature for 220 min. The results are shown in Figure 9. for 70.9 µm side-polished fiber. It is observed that the sensor shows an excellent stability for the temperature response. The variation of wavelength value and power is less than 0.03 nm and 1.2 dB.

The performance between the proposed isopropanol-sealed side-polished fiber in FRL and other temperature FRL sensing system in previous works is compared in Table 1. Here, our proposed temperature sensor shows higher sensitivity compared with other temperature sensors.

## 4. Conclusions

In summary, a compact side-polished fiber temperature sensor filled with isopropanol is developed in FRL. The sensing mechanism of the side-polished fiber and its sensing performance are analyzed. The comparative output wavelength of the FRL has an excellent linearity with the ambient temperature in a range of 25–35 °C. The temperature sensitivity of 70.9 µm side-polished fiber is obtained as ~−2 nm/°C. The sensor has a narrow 3-dB bandwidth of less than 0.06 nm and high SNR of about 60 dB. Moreover, due to the highly narrow bandwidth, the proposed sensor scheme is very suitable for remote monitoring of ambient temperatures. This present side-polished fiber filled with isopropanol in FRL can be a competitive candidate for temperature sensing applications due to its high sensitivity, cost-effective configuration, simple fabrication process and strong mechanical ability.

## Figures and Tables

**Figure 1 micromachines-12-00586-f001:**
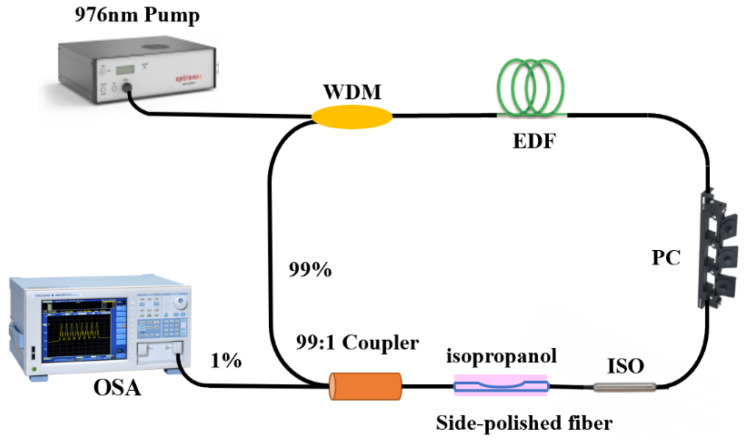
Experiment setup of the temperature FRL sensing system.

**Figure 2 micromachines-12-00586-f002:**
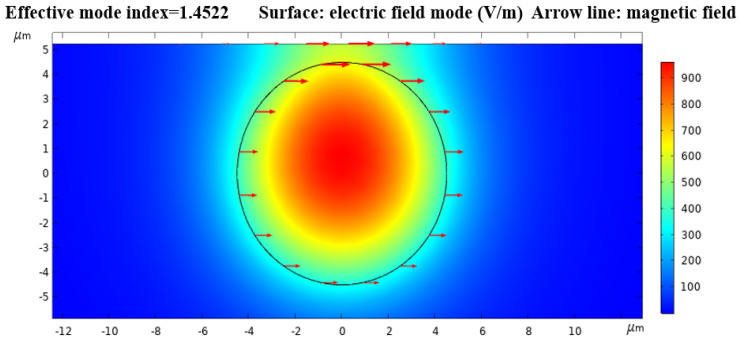
The electric and magnetic transmission field simulation in COMSOL.

**Figure 3 micromachines-12-00586-f003:**
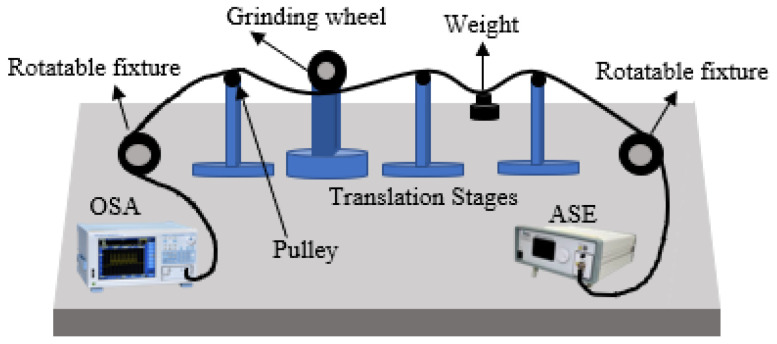
Fiber side-polished system diagram.

**Figure 4 micromachines-12-00586-f004:**
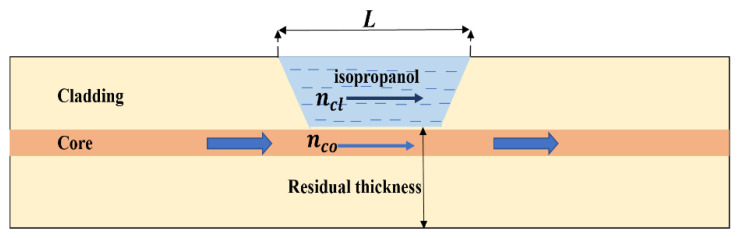
Schematic diagram for the proposed side-polished fiber filled with isopropanol.

**Figure 5 micromachines-12-00586-f005:**
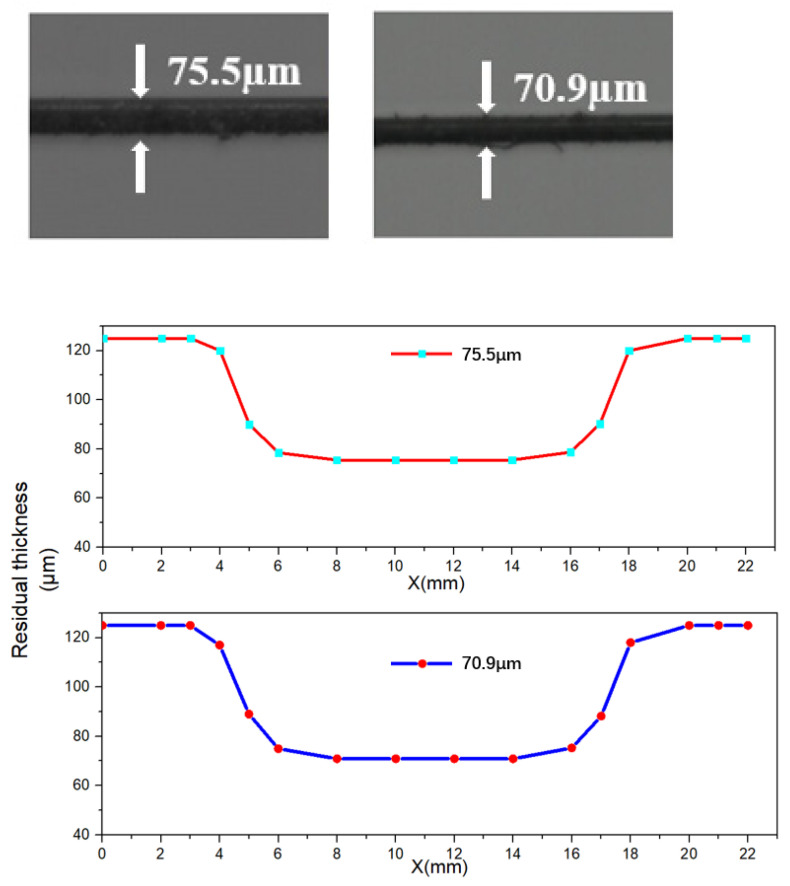
Side view microscopic images of the RT of Side-polished fibers. The RTs were measured as 75.5 µm and 70.9 µm, respectively and Variations of residual thicknesses for the two fabricated side-polished fibers.

**Figure 6 micromachines-12-00586-f006:**
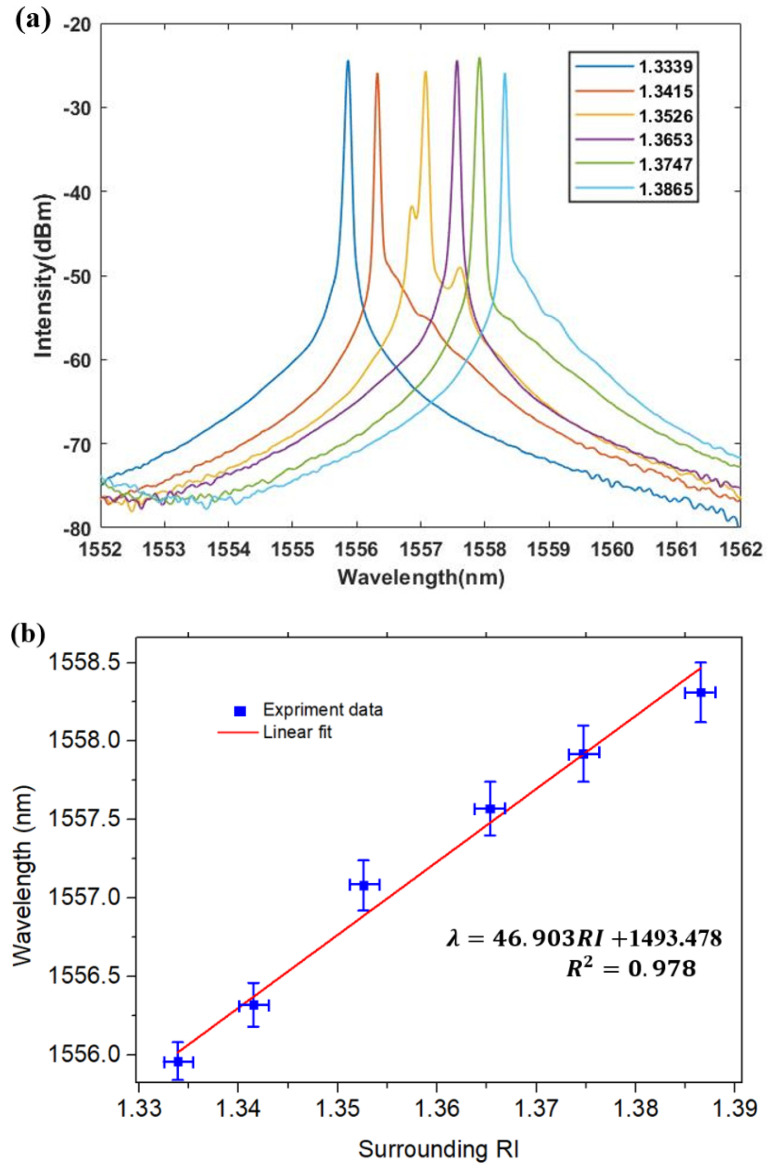
(**a**) Measured output spectra of the RI from 1.3339 to 1.3865 for 75.5 µm side-polished fiber; (**b**) Measured Linear relationship of the RI from 1.3339 to 1.3865 for 75.5 µm side-polished fiber.

**Figure 7 micromachines-12-00586-f007:**
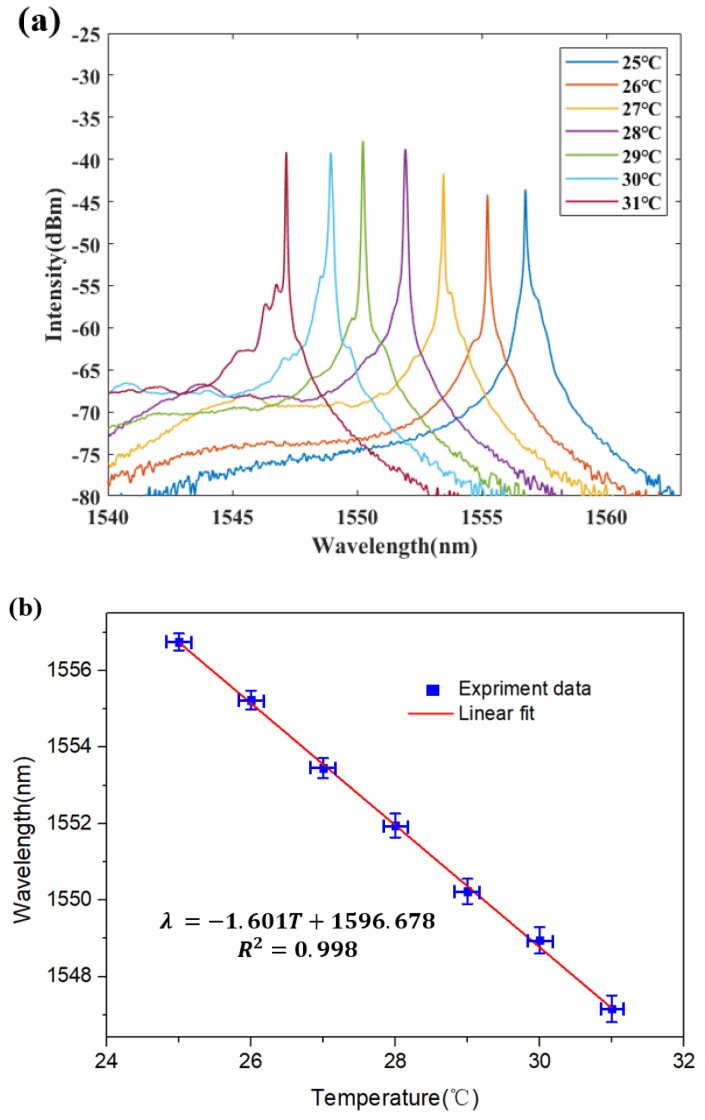
(**a**) Measured output spectra of the 75.5 µm side-polished fiber temperature sensor at various temperatures; (**b**) Linear relationship of wavelength spacing for 75.5 µm side polished fiber.

**Figure 8 micromachines-12-00586-f008:**
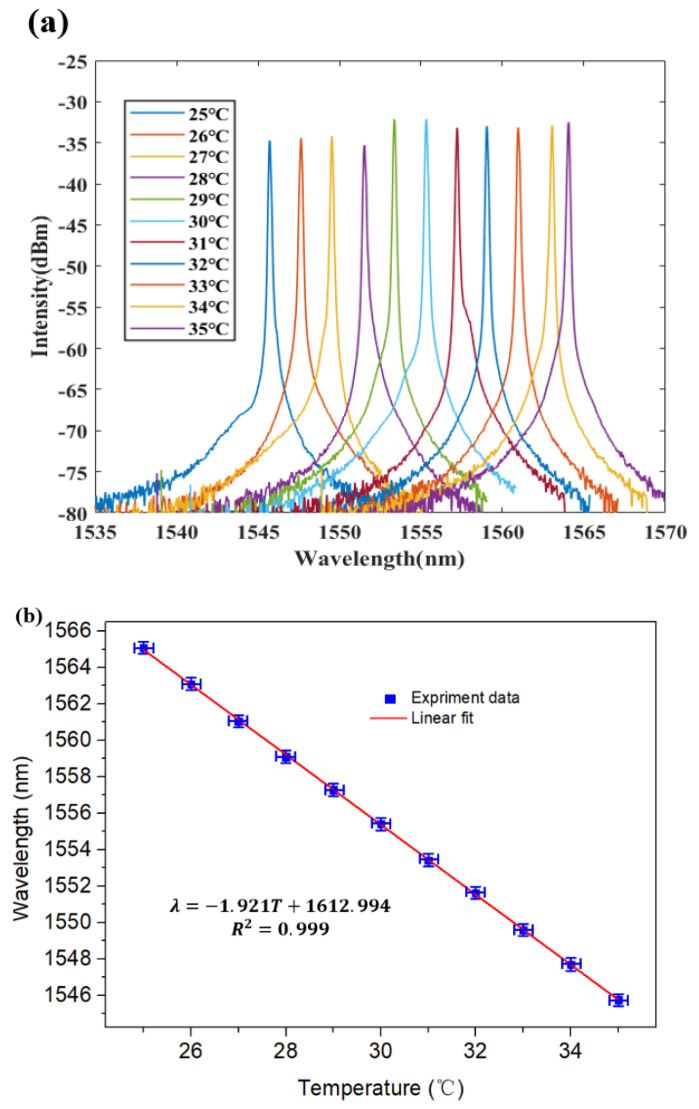
(**a**) Measured output spectra of the 70.9 µm side-polished fiber temperature sensor at various temperature. (**b**) Linear relationship of wavelength spacing for 70.9 µm side-polished fiber.

**Figure 9 micromachines-12-00586-f009:**
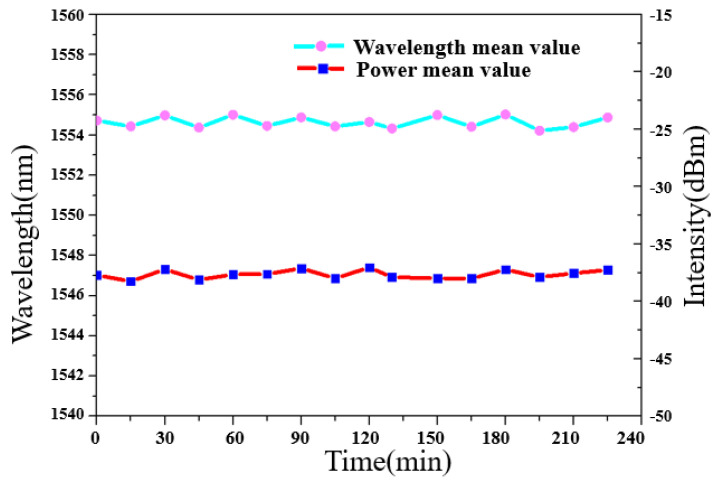
Test for time stability of wavelength shift and power fluctuation.

**Table 1 micromachines-12-00586-t001:** Comparison with other FRL sensing structures.

Strucutres	Sensitivity	Refs.
Three-core fiber + LPFG	47 pm/°C	[33]
Dual-tapered SCF	53.2 pm/°C	[34]
All-solid photonic bandgap fiber	50.9 pm/°C	[35]
Polyimide FBG in fiber ring laser	9.6 pm/°C	[36]
Current work	−1.92 nm/°C	-

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
