# Peer review of "A Fiber Ring Laser Sensor with a Side Polished Evanescent Enhanced Fiber for Highly Sensitive Temperature Measurement"

_micromachines, 2021, doi:10.3390/mi12050586_

Round 1
Reviewer 1 Report
The paper presents a side-polished fibre immersed in isopropanol for temperature sensing. The achieved sensitivities are good, but the theoretical framework and thus the explanation of the results are not convincing. The claimed simplicity is also questionable for a setup containing an EDFA and all components necessary for a ring fibre laser.
I suggest major revisions, which must be quite extensive in order to bring this contribution into a form that is acceptable for publication. Below, I listed some of the most important issues that appeared when I read the manuscript:
- The abstract is hardly understandable. It mentions a lot of details that can be known and understood before reading the paper. Here, the main contribution should be described and explained how the improvement was obtained.
- The introduction mentions very many applications of sensors in the beginning, which are not really specific to the state of the art. The discussion on similar works using side-polished fibres and for fibre-based temperature sensors, however, remains quite short and superficial.
- The theory part seems to be quite independent from the actual measurements and obtained results. The theory is never used throughout the manuscript. I wonder why the authors have actually introduced it. I am not even sure whether the results really originate from an interference of core and cladding modes. The results may also be explicable by the mere change of the effective refractive index in the sensing region due to the thermooptical coefficient of the immersing material. A study on the effect — e.g. by changing the state of polarisation and investigating the polarisation dependence, the influence of the depth of the cladding etc — would help to understand the effect and how to optimise the system.
- The results are shown for two prepared fibres with different sizes. There is no explanation why these are chosen and whether any kind of optimisation has been carried out.
- Fig. 4 shows the spectra of the laser source and the super-continuum source. They are hard to understand, a super-continuum source has not been introduced before, the laser source should be of much narrower spectrum. What are these two spectra supposed to prove?
- The remaining the results just show how the maximum wavelength of the emission spectrum moves to shorter wavelengths with temperature. There are no evaluations of dips or anything, which are discussed in the theory section. How is the maximum and the shift of the maximum to be explained? It just seems that the isopropanol expands and decreases in its refractive index with increasing temperature.
- How do the authors explain the rather high sensitivity of the setup, and how could it be improved? What are the influencing parameters? Depth of polishing, refractive index of core and cladding, the refractive index of the immersing liquid etc.?
Author Response
Re: manuscript Micromachines (ISSN 2072-666X), “Fiber Ring Laser Sensor with Side polished Evanescent enhanced Fiber for highly sensitive temperature measurement”, by Weihao Lin, Yibin Liu , Liyang Shao
Dear Reviewer,
Thank you for relating your comments on our manuscript. We agreed with your comments and helpful suggestions and have made the following revisions accordingly.
Revisions according to comments made by Reviewer 1:
The paper presents a side-polished fibre immersed in isopropanol for temperature sensing. The achieved sensitivities are good, but the theoretical framework and thus the explanation of the results are not convincing. The claimed simplicity is also questionable for a setup containing an EDFA and all components necessary for a ring fibre laser.
I suggest major revisions, which must be quite extensive in order to bring this contribution into a form that is acceptable for publication. Below, I listed some of the most important issues that appeared when I read the manuscript:
- Comment 1: ‘The abstract is hardly understandable. It mentions a lot of details that can be known and understood before reading the paper. Here, the main contribution should be described and explained how the improvement was obtained?’
Reply: Thanks for the reviewer’s comment. The state of the art of the paper has been changed accordingly, few more recent contributions have been described and explained for optical fiber sensors for temperature measurement in detail in the abstract part [underlined part of the introduction].
Abstract part:
We demonstrate a highly sensitive and practical fiber-based temperature sensor. The sensor is constructed based on fiber ring laser (FRL) as well as side-polished fiber filled with isopropanol. The laser cavity of the sensing part fiber is polished by the wheel fiber polishing system with residual thickness (RT) is selected to detect the temperature in the FRL. Thanks to the high thermo-optic coefficient of isopropanol, the sensitivity of the proposed temperature sensor could be effectively improved by filling isopropanol in the cost-less side polished single mode fiber. Refractive index (RI) of isopropanol changes with the surrounding temperature variation allowing high-sensitivity temperature sensing. Experimental results demonstrate that the side polished fiber can efficiently excite high-order cladding modes which enhance the modular interference increase the interaction between the evanescent wave and the isopropanol. Besides, the results show that the sensitivity can be as high as 2nm/°C in the temperature range of 25°C–35°C.
Page 1
Comment 2. “The introduction mentions very many applications of sensors in the beginning, which are not really specific to the state of the art. The discussion on similar works using side-polished fibres and for fibre-based temperature sensors, however, remains quite short and superficial.”
Reply: This is a really a thoughtful comment from the reviewer, we appreciate the reviewer comment.
few more recent contributions of the side-polished fibres and for fibre-based temperature sensors have been described in detail in the abstract part [underlined part of the introduction].
In recent years, optical fiber sensors have attracted increased interest in astronavigation [1], environmental science [2], biomedical monitor [3] and so on, since its geometry has many adaptabilities. Besides, it is suitable for high voltage, electrical noise, high temperature, corrosion and harsh environment [4]. In addition, the device can be made to sense various kinds of physical information [5]. Among them, fiber temperature sensors have been widely explored due to the acceleration of industrial modernization and the rapid development of electronic information industry. Different kinds of optic fiber temperature sensors have been proposed i.e., Fiber Bragg Grating [6], Mach–Zehnder interferometer (MZI) [7], Fabry–Perot interferometer (FPI) [8]. However, the fabrication and application of these extra sensing structure may enhance the deterioration and reduce the signal-to-noise ratio. Wang proposed a response time of optical microfiber temperature sensor based on the microfiber modal interferometer with Sensitivity of -406pm/°C [9]. Lei et al, demonstrated an optical fiber temperature sensor based on a dual-loop optoelectronic oscillator (OEO) with the Vernier effect by slightly different lengths of single mode fiber [10]. Besides, Various structures of optical fibers are designed for temperature measurement [11-15].
Recently, side polished types of fiber sensors have fascinated people’s attention [16-18]. It liberates the need for additional fiber filters. Pei et al. used full vector finite element method to design a D-shaped double clad fiber temperature sensor based on surface plasmon resonance (SPR) with sensitivity of -3.635 nm/°C [19]. However, a gold film is needed to coat the fiber. It greatly increases the cost. Jian et al. proposed a novel fiber optic magnetic field sensor based on D-shaped fiber optic modal interferometer filled with magnetic fluid [20]. The magnetic field sensitivity is 99.68 pm/Oe and the temperature sensitivity is -77.49 pm/℃. The performance of these sensors in terms of temperature is typically limited to ~100pm/℃ and the further improvement is desired. In order to meet the requirements of high signal-to-noise ratio and narrow bandwidth, optical fiber ring laser sensor [21-23] has been widely used in recent years. Besides, benefitted from side-polished optical fiber structure, in 2010, A temperature sensor based on a side-polished fiber (SPF) coupled to a tapered multimode overlay waveguide (MMOW) is proposed by Prerana [24]. Chen et al. designed novel fiber structure in 2018, which coreless side-polished fiber that is wrapped by polydimethylsiloxane [25], a high temperature sensitivity of −0.4409 nm/°C is achieved. Wang el al, demonstrated a surface plasmon resonance (SPR) of a side-polished single mode fiber for temperature sensing thanks to the high thermo-optic and thermal expansion coefficient of the adhesive [26]. In addition, many works based on optical fiber temperature sensor have been reported. [27-31].
New Ref:
- Wen, J.-H.; Wang, J.; Yang, L.; Hou, Y.-F.; Huo, D.-H.; Cai, E.-L.; Xiao, Y.-X.; Wang, S.-S., Response Time of Microfiber Temperature Sensor in Liquid Environment. IEEE Sensors Journal 2020, 20 (12), 6400-6407.
- Cheng, Y.; Wang, Y.; Song, Z.; Lei, J., High-sensitivity optical fiber temperature sensor based on a dual-loop optoelectronic oscillator with the Vernier effect. Opt Express 2020, 28 (23), 35264-35271
- Ren, X.; Gao, J.; Shi, H.; Huang, L.; Zhao, S.; Xu, S., A highly sensitive all-fiber temperature sensor based on the enhanced green upconversion luminescence in Lu2MoO6:Er3+/Yb3+ phosphors by co-doping Li+ ions. Optik 2021, 227.
- Nan, T.; Liu, B.; Wu, Y.; Wang, J.; Mao, Y.; Zhao, L.; Sun, T.; Wang, J., Optical fiber temperature sensor with insensitive refractive index and strain based on phase demodulation. Microwave and Optical Technology Letters 2020, 62 (12), 3733-3738
- Xu, Y.; Chen, X.; Zhu, Y., High Sensitive Temperature Sensor Using a Liquid-core Optical Fiber with Small Refractive Index Difference Between Core and Cladding Materials. Sensors (Basel) 2008, 8 (3), 1872-1878.
- Xu, B.; Li, J.; Li, Y.; Xie, J.; Dong, X., Ultrasensitive Temperature Sensor Based on Refractive Index Liquid-Sealed Thin-Core Fiber Modal Interferometers. IEEE Sensors Journal 2014, 14 (4), 1179-1184.
- Ma, J.; Wu, S.; Cheng, H.; Yang, X.; Wang, S.; Lu, P., Sensitivity-enhanced temperature sensor based on encapsulated S-taper fiber Modal interferometer. Optics & Laser Technology 2021, 139.
- Prerana, P.; Varshney, R. K.; Pal, B. P.; Nagaraju, B., High Sensitive Fiber Optic Temperature Sensor Based on a Side-polished Single-mode Fiber Coupled to a Tapered Multimode Overlay Waveguide. Journal of the Optical Society of Korea 2010, 14 (4), 337-341.
- He, C.; Fang, J.; Zhang, Y.; Yang, Y.; Yu, J.; Zhang, J.; Guan, H.; Qiu, W.; Wu, P.; Dong, J.; Lu, H.; Tang, J.; Zhu, W.; Arsad, N.; Xiao, Y.; Chen, Z., High performance all-fiber temperature sensor based on coreless side-polished fiber wrapped with polydimethylsiloxane. Opt Express 2018, 26 (8), 9686-9699.
- Liu, S.; Cao, S.; Zhang, Z.; Wang, Y.; Liao, C.; Wang, Y., Temperature Sensor Based on Side-Polished Fiber SPR Device Coated with Polymer. Sensors (Basel) 2019, 19 (19).
- Weng, S.; Pei, L.; Wang, J.; Ning, T.; Li, J., High sensitivity D-shaped hole fiber temperature sensor based on surface plasmon resonance with liquid filling. Photonics Research 2017, 5 (2).
- An, G.; Li, S.; Qin, W.; Zhang, W.; Fan, Z.; Bao, Y., High-Sensitivity Refractive Index Sensor Based on D-Shaped Photonic Crystal Fiber with Rectangular Lattice and Nanoscale Gold Film. Plasmonics 2014, 9 (6), 1355-1360.
- Chen, A.; Yu, Z.; Dai, B.; Li, Y., Highly Sensitive Detection of Refractive Index and Temperature Based on Liquid-Filled D-Shape PCF. IEEE Photonics Technology Letters 2021, 33 (11), 529-532
- Dong, Y.; Xiao, S.; Xiao, H.; Liu, J.; Sun, C.; Jian, S., An Optical Liquid-Level Sensor Based on D-Shape Fiber Modal Interferometer. IEEE Photonics Technology Letters 2017, 29 (13), 1067-1070.
- Tan, R. X.; Ho, D.; Tse, C. H.; Tan, Y. C.; Yoo, S. W.; Tjin, S. C.; Ibsen, M., Birefringent Bragg Grating in C-Shaped Optical Fiber as a Temperature-Insensitive Refractometer. Sensors (Basel) 2018, 18 (10).
Comment 3. “The theory part seems to be quite independent from the actual measurements and obtained results. The theory is never used throughout the manuscript. I wonder why the authors have introduced it. I am not even sure whether the results really originate from an interference of core and cladding modes. The results may also be explicable by the mere change of the effective refractive index in the sensing region due to the thermo-optical coefficient of the immersing material. A study on the effect — e.g., by changing the state of polarization and investigating the polarization dependence, the influence of the depth of the cladding etc — would help to understand the effect and how to optimise the system.”
Reply: We thank reviewer for his suggestions, now generation of the mere change of the effective refractive index in the sensing region due to the thermo-optical coefficient of the immersing material has been described elaborately in the theoretical part of the paper. A brief mechanism principle of the sensors is given in the theoretical part.
Theoretical part:
Figure 2 illustrates the schematic diagram of the proposed side polished fiber filled with isopropanol configuration, and the inset shows the cross-sectional view of the side polished fiber employed in the experiment. When the incident light transmitting through the single mode fiber with a fundamental mode reach the side-polished region, leading mismatch between core diameter. Hence, the incident light from the incident fiber is split into two parts by the first polishing point, denoted by and . The interference happens when the two beams recombine at the core of the fiber, and the resultant interference light intensity can be mathematically described by
(1)
Which and delegate the light intensities of beams transmitting through the residual core and the side-opened fiber, respectively.
Figure 1. Experiment setup of the temperature FRL sensing system.
The evanescent wave in cladding mode can be enhanced effectively by polishing the fiber side. In the light of the optical transmission process, the refractive index of the cladding modes and core mode rely on the temperature(T) for the thermal-optical coefficient of fiber. The equation could be further simplified as [19-20]:
(2)
Which is the refractive index of the core mode with respect to the temperature change and (m =1,2,3…) is the refractive index of the mth-cladding modes with respect to the temperature change, is the dissimilarity between the cladding modes and core mode, L represents the length of side-polished fiber and λ is the center wavelength of input pump source which is 976nm.The fringe visibility of the interferogram is as follows[20]:
Equation (3) shows that the visibility of fringes of interference pattern is mainly determined by the intensity of the core mode and the cavity mode, that is, the smaller the intensity difference between the core mode and the air mode in the grinding and polishing area, the higher the visibility of fringes of interference pattern. In the structure of optical fiber based on measurement and polishing, the output region is used as the input coupler of mode excitation, which means that the lateral offset plays an important role in the performance of MZI. In addition, the different grinding depth loss and transmission loss of the core mode and cavity mode also affect the visibility of the fringes. Therefore, we use COMSOL method to study the waveguide electric field transmission characteristics at 1550 nm wavelength, and the results are shown in Fig. 2. Some energy transmitted outside the fiber core.
Figure. 2. (a) The electric and magnetic transmission field simulation in COMSOL.
At the end of the introduced fiber, most of the light intensity is concentrated in the core of SMF, and a small amount of light intensity is distributed in the cladding of SMF. If the transverse offset is introduced at the first splice point, the core mode and cavity mode will be excited simultaneously when the incident light is coupled to the polished fiber. The larger the lateral offset is, the more energy is coupled into the cavity mode. This makes it possible to obtain MZI with excellent performance according to equation (3), which represents a very clear interference spectrum with high signal-to-noise ratio and fringe visibility
Equation (2) shows that the phase difference between the core mode and the cavity mode strongly depends on their effective RI difference and the length of the interference region. If l is a constant, the temperature sensitivity of MZI can be expressed by formula (2)[19-20]:
where and are the thermo-optic coefficients of the RI of fiber core and isopropanol, respectively. The core of the single mode fiber's thermo-optic coefficient d ∼8.6 × 10−6∕°C. While isopropanol with high thermo-optic coefficient of ∼ − 4.5 × 10−4∕°C is infiltrated into the side polished fiber, the core mode and the isopropanol mode vary quickly and differently when the temperature changes. Besides, since the refractive index value of isopropanol changes with the temperature [20], the effective RI difference between the core mode and the cavity mode will enhance with the increment of temperature.
Page 3-5
Comment 4. “The results are shown for two prepared fibers with different sizes. There is no explanation why these are chosen and whether any kind of optimization has been carried out.”
Reply: Authors thank for the reviewer comments; it is very significant. Since the core of the single mode fiber's thermo-optic coefficient d ∼8.6 × 10−6∕°C. While isopropanol with high thermo-optic coefficient of ∼ − 4.5 × 10−4∕°C is infiltrated into the side polished fiber, the core mode and the isopropanol mode vary quickly and differently when the temperature changes. Besides, since the refractive index value of isopropanol changes with the temperature, the effective RI difference between the core mode and the cavity mode will enhance with the increment of temperature. More importantly, the air-cavity is closer to the core, the interaction with core mode and surrounding RI is more sensitivity to temperature, besides, to protect the fiber from being destroyed a waist of 75.5 µm and 70.9 µm are selected for temperature sensing.
Page 6
Comment 5 “Fig. 4 shows the spectra of the laser source and the super-continuum source. They are hard to understand, a super-continuum source has not been introduced before, the laser source should be of much narrower spectrum. What are these two spectra supposed to prove?”
Reply: We appreciate the observations of the reviewer. Truly speaking it is not necessary for proposed the figure 4 to show the spectra of the laser source and the super-continuum source, here, we delete the figure 4 to avoid misleading the reviewer. We replace the figure 4 into figure3 and figure5 to explain the polished process. In this paper our aim is just to show the possibility of the sensor as a meaningful temperature sensing application in fiber ring laser system. The output of the super-continuum source is the result of figure 3 during the polishing process. Figure 5 is used to in distinct the residual thickness of the fiber sensing system.
Figure 3. Fiber side-polished system diagram.
Figure 4.Side view microscopic images of the RT of Side-polished fibers. The RTs were measured as 75.5 µm and 70.9 µm, respectively
Page 6&7
Comment 6: “The remaining the results just show how the maximum wavelength of the emission spectrum moves to shorter wavelengths with temperature. There are no evaluations of dips or anything, which are discussed in the theory section. How is the maximum and the shift of the maximum to be explained? It just seems that the isopropanol expands and decreases in its refractive index with increasing temperature.”
Reply: We thank the reviewer for his remarks, to be honest there are no evaluations of dips, hence we change the theory part accordingly. The theoretically part is explained directly in reply to comment 2. As shown in above.
Theoretically part:
When the incident light transmitting through the single mode fiber with a fundamental mode reach the side-polished region, leading mismatch between core diameter. Hence, the incident light from the incident fiber is split into two parts by the first polishing point, denoted by and . The interference happens when the two beams recombine at the core of the fiber, and the resultant interference light intensity can be mathematically described by
(1)
Which and delegate the light intensities of beams transmitting through the residual core and the side-opened fiber, respectively.
The evanescent wave in cladding mode can be enhanced effectively by polishing the fiber side. In the light of the optical transmission process, the refractive index of the cladding modes and core mode rely on the temperature(T) for the thermal-optical coefficient of fiber. The equation could be further simplified as [19-20]:
Which is the refractive index of the core mode with respect to the temperature change and (m =1,2,3…) is the refractive index of the mth-cladding modes with respect to the temperature change, is the dissimilarity between the cladding modes and core mode, L represents the length of side-polished fiber and λ is the center wavelength of input pump source which is 976nm.The fringe visibility of the interferogram is as follows[20]:
Equation (3) shows that the visibility of fringes of interference pattern is mainly determined by the intensity of the core mode and the cavity mode, that is, the smaller the intensity difference between the core mode and the air mode in the grinding and polishing area, the higher the visibility of fringes of interference pattern. In the structure of optical fiber based on measurement and polishing, the output region is used as the input coupler of mode excitation, which means that the lateral offset plays an important role in the performance of MZI. In addition, the different grinding depth loss and transmission loss of the core mode and cavity mode also affect the visibility of the fringes.
At the end of the introduced fiber, most of the light intensity is concentrated in the core of SMF, and a small amount of light intensity is distributed in the cladding of SMF. If the transverse offset is introduced at the first splice point, the core mode and cavity mode will be excited simultaneously when the incident light is coupled to the polished fiber. The larger the lateral offset is, the more energy is coupled into the cavity mode. This makes it possible to obtain MZI with excellent performance according to equation (3), which represents a very clear interference spectrum with high signal-to-noise ratio and fringe visibility
Equation (2) shows that the phase difference between the core mode and the cavity mode strongly depends on their effective RI difference and the length of the interference region. If l is a constant, the temperature sensitivity of MZI can be expressed by formula (2)[19-20]:
where and are the thermo-optic coefficients of the RI of fiber core and isopropanol, respectively. The core of the single mode fiber's thermo-optic coefficient d ∼8.6 × 10−6∕°C. While isopropanol with high thermo-optic coefficient of ∼ − 4.5 × 10−4∕°C is infiltrated into the side polished fiber, the core mode and the isopropanol mode vary quickly and differently when the temperature changes. Besides, since the refractive index value of isopropanol changes with the temperature [20], the effective RI difference between the core mode and the cavity mode will enhance with the increment of temperature.
Page 3-5
Comment 7: “How do the authors explain the rather high sensitivity of the setup, and how could it be improved? What are the influencing parameters? Depth of polishing, refractive index of core and cladding, the refractive index of the immersing liquid etc.?”
Reply: As mentioned above that the phase difference between the core mode and the air-cavity mode is strongly dependent on their effective RI difference, as well as the length of the interferometer region. Assuming that L is a constant, the temperature sensitivity of the MZI can be derived from Eq. (1) a
The core of the SMF is made of pure silica, so its thermooptic coefficient is ∼8.6 × 10−6∕°C. When liquid with high thermo-optic coefficient (e.g., isopropanol with a thermo-optic coefficient of ∼ − 4.5 × 10−4∕°C) is infiltrated into the air cavity, the core mode and the air-cavity mode vary quickly and differently when the temperature varies. In addition, because the RI value of isopropanol changes in the range of 1.3766–1.3631 in the temperature range of 20°C–50°C [1], the effective RI difference between the core mode and the air-cavity mode will increase with the increment of temperature and cause the transmission spectra to redshift. Note that higher temperature sensitivity and a wider measurement range can be acquired if we infiltrate some liquid having a higher thermo-optic coefficient and higher boiling point into the air cavity.
New Ref:
[1] S. J. Qiu, Y. Chen, F. Xu, and Y. Q. Lu, Opt. Lett. 37, 863 (2012).
We thank the reviewer once again for their helpful suggestions and comments, which have improved the quality of our manuscript greatly.
On behalf of all authors,
Yours sincerely,
Li-Yang Shao

Reviewer 2 Report
The proposed paper describes a study on a temperature sensor based on side-polished optical fiber. Presented results are straightforward and easy to interpret. Spectral characteristics of the proposed sensor are presented with linear fits to data obtained for changing external refractive index and temperature.
The manuscript has a lot of grammar errors and a few spelling mistakes, thus a professional English editing of the manuscript is necessary.
My specific comments:
- In Fig. 5 there are shown only 5 spectra, while in Fig. 6 there are 6 points representing consecutive measurements. It Figs. 6 also the 6th missing spectrum should be presented.
- The error bars in Fig. 6, 8 and 10 have different values for different measured points. If all of the measurements were performed in the same way, there is no reason why those bars should have different values not arranged in increasing (or decreasing) order.
- In Figs. 6, 8, 10 there are no x-axis error bars, which should be added.
- The "y" and "x" in the equations presented in Figs. 6, 8 and 10 should be changed to "\lambda" and "RI" / "\degree C" respectively for clarity.
- Reference 1 in Table 1 should be corrected.
- It would be better to reorganize pairs of Figs. 5-6, 7-8 and 9-10 into single figures with two side-by-side panels (a and b).
Author Response
Re: manuscript Micromachines (ISSN 2072-666X), “Fiber Ring Laser Sensor with Side polished Evanescent enhanced Fiber for highly sensitive temperature measurement”, by Weihao Lin, Yibin Liu , Liyang Shao
Dear Reviewer,
Thank you for relating your comments on our manuscript. We agreed with your comments and helpful suggestions and have made the following revisions accordingly.
Revisions according to comments made by Reviewer :
The proposed paper describes a study on a temperature sensor based on side-polished optical fiber. Presented results are straightforward and easy to interpret. Spectral characteristics of the proposed sensor are presented with linear fits to data obtained for changing external refractive index and temperature.
The manuscript has a lot of grammar errors and a few spelling mistakes; thus, a professional English editing of the manuscript is necessary.
Comment 1: ‘In Fig. 5 there are shown only 5 spectra, while in Fig. 6 there are 6 points representing consecutive measurements. It Figs. 6 also the 6th missing spectrum should be presented.’
Reply: Thanks for the reviewer’s comment. The state of the art of the paper has been changed accordingly.
Figure. 5. Measured output spectra of the RI from 1.3339 to 1.3865 for 75.5 µm side-polished
Page 8
Comment 2. “The error bars in Fig. 6, 8 and 10 have different values for different measured points. If all of the measurements were performed in the same way, there is no reason why those bars should have different values not arranged in increasing (or decreasing) order.”
Comment 3. “In Figs. 6, 8, 10 there are no x-axis error bars, which should be added”
Comment 4 “The "y" and "x" in the equations presented in Figs. 6, 8 and 10 should be changed to "\lambda" and "RI" / "\degree C" respectively for clarity.”
Comment 6: It would be better to reorganize pairs of Figs. 5-6, 7-8 and 9-10 into single figures with two side-by-side panels (a and b).
Reply: Those are really a thoughtful comment from the reviewer, we appreciate the reviewer comments. there x-axis error bars have been added. “The "y" and "x" in the equations presented in Figs. 6, 8 and 10 have been changed to "\lambda" and "RI" / "\degree C" respectively for clarity. Figs. 5-6, 7-8 and 9-10 into single figures have been combined with two side-by-side panels (a and b). Besides, here we think the error bars have different values for different measured points may due to we use different test result at the different test times. We now correct the error bar into the same test time and get the same result as different values arranged in increasing order.
Comment 5: Reference 1 in Table 1 should be corrected.
Reply: We appreciate the observations of the reviewer. Reference 1 in Table 1 should be corrected has been corrected accordingly.
Authors appreciate the concerns of the reviewer and also thankful to him for his comments. This total paper has been reviewed and corrected accordingly. Many lines are corrected and changed for ease of understanding
We thank the reviewer once again for their helpful suggestions and comments, which have improved the quality of our manuscript greatly.
On behalf of all authors,
Yours sincerely,
Li-Yang Shao
